# The Work Ability Index (WAI) in the Healthcare Sector: A Cross-Sectional/Retrospective Assessment of the Questionnaire

**DOI:** 10.3390/ijerph21030349

**Published:** 2024-03-15

**Authors:** Nicola Magnavita, Igor Meraglia, Giacomo Viti, Lorenzo Borghese

**Affiliations:** 1Department of Life Sciences and Public Health, Università Cattolica del Sacro Cuore, 00168 Rome, Italy; meraglia.igor93@gmail.com (I.M.); giacomo.viti.med@gmail.com (G.V.); 2S. Giovanni Battista Hospital, Association of Italian Knights of the Sovereign Military Order of Malta, 00148 Rome, Italy; presidente@acismom.it

**Keywords:** disability management, total worker health, psychometrics, medical surveillance, health promotion, musculoskeletal disorders, psychiatric disorders, ageing, gender differences, night work

## Abstract

The Work Ability Index (WAI) is the most widely used questionnaire for the self-assessment of working ability. Because of its different applications, shorter versions, and widespread use in healthcare activities, assessing its characteristics is worthwhile. The WAI was distributed online among the employees of a healthcare company; the results were compared with data contained in the employees’ personal health records and with absence registers. A total of 340 out of 575 workers (59.1%) participated; 6.5% of them reported poor work ability. Exploratory factor analysis indicated that the one-factor version best described the characteristics of the WAI. The scores of the complete WAI, the shorter form without the list of diseases, and the minimal one-item version (WAS) had equal distribution and were significantly correlated. The WAI score was inversely related to age and significantly lower in women than in men, but it was higher in night workers than in their day shift counterparts due to the probable effect of selective factors. The WAI score was also correlated with absenteeism, but no differences were found between males and females in the average number of absences, suggesting that cultural or emotional factors influence the self-rating of the WAI. Workers tended to over-report illnesses in the online survey compared to data collected during occupational health checks. Musculoskeletal disorders were the most frequently reported illnesses (53%). Psychiatric illnesses affected 21% of workers and had the greatest impact on work ability. Multilevel ergonomic and human factor intervention seems to be needed to recover the working capacity of healthcare workers.

## 1. Introduction

The Work Ability Index (WAI) [1] is a questionnaire for the self-assessment of working capacity. It is used in occupational healthcare to reveal how well a worker can perform his or her work. It was developed in the late 1990s by Finnish researchers and is widely used as one of the methods for assessing work ability during medical surveillance and workplace surveys. This tool enables the occupational doctor who is beginning risk prevention activities in a company to rapidly collect information on the occupational health status of workers and subsequently create a well-targeted health plan. Over 97% of Finnish doctors and a substantial share of occupational physicians, epidemiologists, psychologists, and sociologists from all over the world currently use it for health promotion, population health trend analysis, individual case management, and for calculating work ability across age groups [2]. An abundant quantity of literature bears witness to the diffusion of the questionnaire. As of 15 February 2024, searching for the term “Work Ability Index” in the PubMed search engine yielded 5535 articles. The professional category most frequently investigated with the WAI are healthcare workers (HCWs), and a meta-analysis indicates that low working capacity patterns are frequent among these workers [3]. Despite the abundance of contributions in the literature, some important elements concerning the structure and application of the questionnaire still need to be clarified.

The first aspect to clarify concerns the psychometric characteristics of the questionnaire. The authors conceived the questionnaire as a unitary instrument that would provide a single score, while some studies have observed that a two- or three-factor solution could be more advantageous [4]. The second relevant aspect regards the diffusion of shortened versions of the WAI, which could be particularly useful in investigations conducted in the workplace, but which require a careful evaluation of the instrument. The third characteristic needing clarification is related to the observation that the WAI score is influenced by age, gender, and type of job. The last important aspect concerns the level of reliability of self-assessments.

Despite the amount of research conducted with this tool, few critical evaluations have associated the administration of the questionnaire with objective data (e.g., the registration of absences due to illness [5] or clinical assessments of workers [6,7,8]) to verify the correspondence between subjective answers and objective information. This aspect is very important for a company’s occupational doctor, who must take the necessary measures to protect the health of workers and must therefore know whether subjective declarations correspond with objective data. Moreover, although several authors have used shorter versions of the questionnaire (e.g., without a list of diseases or with just one component [9]), there are very few studies that have verified the usefulness of such tools. Lastly, it is also important for the occupational doctor to know whether the questionnaire functions in a particular way in specific work sectors, such as the healthcare field, where the greatest number of applications have been made to date.

The WAI includes a list of diseases and seven occupational health characteristics. The final WAI score is calculated as the total unweighted score covering the WAI’s seven dimensions, or indicators. These are as follows:Current work ability compared with lifetime best;Work ability in relation to the demands of the job;Number of current diseases diagnosed by a physician;Estimated work impairment due to illness;Sick leave during the past 12 months;Personal prognosis of work ability 2 years from now;Mental resources.

The combination of dimension values results in a total WAI score that can range from 7 (unable to work) to 49 (full work ability). Based on this score, an individual is classified into standard work ability categories of excellent (WAI 44–49), good (WAI 37–43), moderate (WAI 28–36), and poor (WAI ~27).

The list of diseases is the aspect of the questionnaire that has posed most problems in its application. A lengthy list of 51 medical problems was included in the original WAI, requiring employees to note any illnesses for which they had received a diagnosis. Filling out the questionnaire was too complicated to perform in the workplace where health surveillance must not interfere with production needs. A shortlist of 15 diseases was included in the WAI version introduced by Nubling et al. [10]. This turned out to be comparable to the extended version and is nowadays commonly used in occupational medicine. Despite this simplification, internal consistency studies have shown that most missing responses on the WAI relate to the list of diseases, and that the coherence of the WAI is not significantly influenced by self-reported symptoms and illnesses [11]. Some researchers have therefore proposed solving the question of ambiguous WAI scores resulting from missing answers by using a version of the WAI that does not include the list of diseases [12]. This shortened version, without the list of diseases (WAInodis), has values ranging from 6 to 42.

An even shorter version of the questionnaire is based on the first question “Assume that your work ability at its best has a value of 10 points. How many points would you give your current work ability?”. This single-item questionnaire, also called Work Ability Score (WAS), has been proposed for use in extensive population surveys [12].

To obtain a more thorough knowledge of the questionnaire, the questions we wanted to answer were as follows:Is the WAI score significantly influenced by the age, gender, or occupational category of the workers?Does the questionnaire have a unifactorial structure, or is it better to recognize two or more distinct factors within it?Does the simplified questionnaire, without the list of diseases, have an acceptable correlation with the complete version?Does the shortest version of the questionnaire (limited to the first dimension) have an acceptable correlation with the final score obtained from the complete questionnaire?Is the assessment of the demands of the job reliable, i.e., do workers who carry out the same job consider it equally demanding from a physical or psychological point of view?Is there a correspondence between the diseases reported in the questionnaire and those reported to the occupational doctor during the medical examination carried out at the workplace?Are illnesses not diagnosed by a doctor, and which therefore do not contribute to the WAI score, irrelevant for the purposes of assessing work capacity?Is there an association between the WAI and sickness absence?Which group of pathologies determines the greatest impairment of the WAI? And which determines the highest burden of sickness absence?Are mental resources correlated with the presence of mental illnesses?

To answer these questions, we administered the WAI to all the employees of a healthcare company who were exposed to occupational risks and compared the answers with the data collected during medical surveillance and in sickness absence records.

## 2. Materials and Methods

### 2.1. Population

In Italy, according to Council Directive 89/391/EEC of 12 June 1989 [13] concerning the implementation of measures aimed at promoting improvements in the safety and health of workers at work and Legislative Decree 81/2008 which incorporated the EEC Directive into national legislation [14], workers who are exposed to occupational risks must be subjected to health surveillance in the workplace. During this surveillance, HCWs from a public hospital service were invited to complete the WAI questionnaire using the SurveyMonkey© online platform. The estimated completion time of the questionnaire was 5 min.

All workers exposed to occupational risks and included in the health surveillance register (575) were confidentially contacted by email and invited to participate in the survey. Participation was completely voluntary, no economic incentive was provided for response, and no sanctions were adopted for non-responders. Participants were enrolled between 1 and 30 November 2023. Two weeks after commencing the investigation, a reminder mail was sent to non-responders.

This research study was conducted in accordance with the Helsinki Declaration. Participants signed an informed consent form and, by signing their personal health file, authorized the analysis of their personal data and the dissemination of the results in a collective anonymous form according to the principles of confidentiality of occupational medicine and the ICOH code of ethics [15]. Ethics approval was obtained from the Catholic University Ethics Committee (ID 3008). Given that this was a survey, the only criterion for admission was to be an active worker registered in the health surveillance registers, while the exclusion criterion was refusal to respond or incomplete response. Given the cross-sectional study design, no imputation was applied for missing data and the results were based on completed survey responses.

### 2.2. Questionnaire and Objective Data

Via the online platform, workers were invited to provide their name, age, gender, occupational category, and to state whether they worked night shifts. They were then invited to self-assess their working capacity using the Italian version of the WAI [16].

The WAI is made up of a series of questions that consider the resources and health status of the worker as well as the physical and mental demands of their job. Answers are evaluated in the seven dimensions mentioned in the Introduction, yielding a score ranging from 7 to 49. The value assigned represents the employee’s perception of their own work capacity and, consequently, their work capacity level.

In addition to the score resulting from the complete WAI questionnaire, we calculated the score of the shortened version (with values ranging from 6 to 42), which does not take into account diseases diagnosed by a doctor (WAInodis), and the shortest version, also known as Work Ability Scale (WAS), that corresponds to the first dimension of the questionnaire, with values ranging from 0 to 10.

For the period from 2021 to October 2023, the healthcare company’s administration provided us with its absence registers. We also analyzed the workers’ personal health files containing the results of medical examinations and health checks conducted in the workplace over the years to evaluate the correspondence between subjective indications and pathologies objectively identified by the occupational doctor.

### 2.3. Statistics

Firstly, using Cronbach’s α, the internal consistency of the WAI and its shortened versions was assessed. Using the Kolmogorov–Smirnov and Shapiro–Wilk tests, the distribution of the variables of interest (WAI, WAInodis, and WAS) was studied to ascertain whether it was normal. To estimate the adequacy of substituting the WAInodis or the WAS for the WAI when assessing work ability, the convergent validity between the two instruments was tested by evaluating the Spearman correlation between the shortened forms and the total WAI score. Then, the construction of a linear regression model enabled us to investigate the concurrent validity of the WAI and its shorter versions. In particular, we looked at whether the scores of the shorter WAI versions predicted the overall WAI score to a reasonable extent.

Exploratory factor analysis, principal component analysis with direct oblimin rotation, and the maximum likelihood technique were used to assess the construct validity of the questionnaire.

The distribution of the WAI score between the different occupational categories was studied using analysis of variance (one-way ANOVA) and Bonferroni post hoc comparisons.

Simple linear regression was used to evaluate the relationship between independent variables (gender, age, job category, night shifts, days of absence) and the WAI score.

IBM/SPSS Statistics for Windows, Version 28.0. (Armonk, NY, USA: IBM Corp.) was used for the analyses.

## 3. Results

An invitation to participate was sent via email to all employees of the healthcare company who were on the occupational hazard watch list (n = 575). A total of 412 workers (71.7% of those eligible) responded to the invitation, but 62 of them were excluded because their responses were incomplete. Of the 340 workers (59.1% of the cohort) who finally took part, 133 were male (39.1%) and 207 were female (60.9%). The mean age was 46.3 ± 11.6 years.

The average score of the WAI questionnaire was equal to 37.0 ± 5.6 points; median value 38. The distribution of scores was significantly non-normal (Kolmogorov–Smirnov test 0.105, *p* < 0.001; Shapiro–Wilk test 0.951, *p* < 0.001) (Figure 1). The reliability coefficient (Cronbach’s α) of the questionnaire was 0.73.

In accordance with the authors’ indications [1,17], we classified the workers by dividing them into four groups of increasing ability based on the WAI score. A total of 22 workers (6.5%) reported poor work ability (score 7–27); 119 (35.0%) had moderate work ability (score 28–36); 157 (46.2%) reported good work ability (score 37–43); and 42 (12.4%) showed excellent work ability (score 44–49).

### 3.1. Distribution of Work Ability by Age, Gender, and Occupational Category

The WAI was inversely correlated with age (Spearman’s rho = −0.294 *p* < 0.001). The WAI score decreased by approximately one point every three years of age.

Female workers had significantly worse WAI scores than male ones (36.3 ± 6.0 in female vs. 37.9 ± 4.9 in male workers, Mann–Whitney U Test *p* < 0.05).

The distribution of the WAI score varied significantly between the different occupational categories (ANOVA *p* < 0.001). Nurses reported the lowest work ability, while assistant nurses had the highest average rating of work ability. The Bonferroni test showed that there was a significant difference between nurses and assistant nurses (*p* < 0.001), between nurses and clerks (*p* < 0.01), and between assistant nurses and technicians (*p* < 0.05) (Table 1).

Workers performing night shifts (n = 90) reported a significantly better WAI score than others (38.2 ± 5.8 vs. 36.5 ± 5.5, *p* < 0.01). Night workers were significantly younger than other workers (42.4 ± 9.9 vs. 47.7 ± 11.9 years, Student’s *t* test *p* < 0.001) and included a larger proportion of nurses. Limiting the comparison to nurses and assistant nurses (n = 172 workers), night workers had a better WAI than those who did not perform night shifts (37.9 ± 6.0 vs. 35.8 ± 6.2, *p* < 0.05) and were also younger (42.3 ± 9.3 vs. 48.9 ± 10.9 years, Student’s *t* test *p* < 0.001).

### 3.2. Factorial Structure of the Work Ability Index

Principal component analysis suggested that the presence of a single factor explained over 42% of the variance. All items showed acceptable loadings (>0.49) within the model. The least variance was explained by item 5, which accounted for 49% of variance, whilst the most explanatory item was item 2 (74%). An alternative two-factor model was examined to provide further assurance that the one-factor structure was the most appropriate. The two-factor model explained over 55% of the variance. In the two-factor model, the first factor was associated with six indicators, while the second factor (which explained 13% of the variance) was associated with sickness absences (Table 2).

### 3.3. Relationships between the Complete WAI and the Shortened Version

The shortened version of the questionnaire without the list of diseases (WAInodis) had a similar distribution to that of the total score (Figure 2). The two scores were strongly correlated with each other (Spearman’s rho = 0.96 *p* < 0.001) (Table 3). Using simple linear regression to calculate the WAI score and taking the WAInodis score as a predictor, we obtained a standardized coefficient beta = 0.962 and a coefficient of determination (R square) of 0.93.

### 3.4. Relationships between the Complete WAI and the One-Item Version

The first dimension of the WAI, or the Work Ability Score (WAS), demonstrating current working capacity compared to lifetime best, produced a similar distribution to the total score obtained from the seven indicators (Figure 3). To interpret WAS results, the inventors of this method [18] suggested using the same type of categorization as for the WAI, i.e., poor (0–5 points), moderate (6, 7), good (8, 9), excellent (10).

The correlation between the WAS and the WAI was highly significant (Table 3). In a simple linear regression model, the WAS, inserted as a predictor of the WAI, yielded a standardized coefficient beta = 0.752, with a coefficient of determination (R square) of 0.57.

### 3.5. Reliability of the Question on Job Characteristics

The second indicator of the WAI, “Work ability in relation to the demands of the job”, is based on three questions: the first asks whether the job is psychologically demanding, physically demanding, or both; the other two questions ask workers to evaluate the physical and psychological commitment required by the job. Depending on the answer to the first question, the scores obtained from the last two questions must be corrected, either by increasing them by 50% or by reducing them by 50%. It is therefore important to check whether the answer to the first question is consistent with the work profile of the respondents.

In our sample, most workers (64.4%) described their job as psychologically and physically demanding; 30.6% believed it was psychologically demanding. Only 5% (three nurses, three physiotherapists, and eleven assistant nurses) indicated that the work was physically demanding but without any psychological burden. The tasks assigned to the aforementioned workers were not unlike those performed by other members of the same occupational category.

### 3.6. Reliability of the Reported Diseases

The third indicator of the WAI is “Number of current diseases diagnosed by a physician”. The availability of health records made it possible to compare WAI responses with data collected in personal health files.

First, we verified that workers who reported suffering from one or more illnesses had declared these conditions during medical examinations carried out in the workplace to assess work fitness. In most cases, the reported disease groups roughly corresponded with the pathologies reported to the occupational doctor. However, we noticed that many workers indicated a greater number of classes of disorders than declared in the routine health checks they undergo to establish fitness for work. The tendency not to report illnesses during routine medical examinations seemed to depend on the type of illness. Although musculoskeletal disorders were almost always reported during health surveillance, respondents reported other types of diseases less frequently. For example, cancers had a fair degree of under-reporting. In this sample, four workers reported a benign or malignant tumor diagnosed by the doctor, and twelve workers declared that “in their opinion” they suffered from benign or malignant tumors. In the medical records, we found a tumor history in seven cases (two breast tumors, one thyroid tumor, a neurofibroma, a mastocytosis, a giant cell tumor, and a lymphoma), but no reports of neoplasia in the remaining five. The workers who had reported the presence of tumors were all declared fit for work, without any limitations.

Another aspect that we were able to compare with the health record data was the first dimension, concerning current work capacity compared to maximum work capacity (WAS). Although five workers declared that they were completely unable to work (WAS = 0), their health records enabled us to ascertain that they did not have serious chronic diseases and had never expressed problems in carrying out work tasks in the past. A more thorough investigation, conducted by examining the absence registers and questioning the workers, revealed that they had completed the questionnaire while they were on sick leave but had regularly resumed work after recovering from illness.

Finally, we analyzed the situation of the 22 workers with a WAI score < 27. For six of them, the occupational doctor had expressed limited fitness for work, while all the others had been declared fit for work.

### 3.7. Relevance of Diseases Not Confirmed by a Doctor

In the disease section, the WAI questionnaire provides three response options, relating respectively to absence of disease, the presence of self-diagnosed disease, and disease certified by a doctor. We wanted to evaluate whether there was a difference in work capacity between subjects who reported illnesses and colleagues. To make comparisons without being influenced by the number of diseases reported, we used the WAInodis score (Table 4). The ANOVA (analysis of variance) and post hoc comparisons using the Bonferroni test showed that the difference in scores was generally significant between the absence and presence of diseases, regardless of whether they had been diagnosed by a doctor or not. Across nearly all disease groups, there was no difference in work ability levels in doctor-diagnosed cases compared to workers’ self-diagnosed cases. Only musculoskeletal disorders were associated with worse work ability in cases diagnosed and treated by a doctor, compared to self-diagnosed disorders. In four disease groups (endocrine and metabolic diseases, diseases of the blood, abortions or neonatal defects in children, and tumors), no significant difference in WAInodis was found between healthy and sick people.

### 3.8. Association between Diseases and Impairment

An analysis of the questionnaire answers enabled us to evaluate the prevalence of different groups of diseases. Most workers (180 out of 340, 53%) reported being affected by musculoskeletal disorders in the back, neck, limbs, or other parts of the body (e.g., neck pain, back pain, sciatica, recurrent pain, rheumatism, arthritis). The third category of problems in order of frequency, but the first in order of impairment, was the presence of mental health disorders (depression, anxiety, sleep problems, and burnout), which were reported by 73 respondents (21.5%).

In terms of frequency and impact, other very important illnesses were neurological and sensory diseases (migraine, epilepsy, stroke, neuralgia, eye and hearing diseases, other neurological diseases, 20.3%), gastroenteric diseases (gastritis, gastroduodenal ulcer, gallbladder stones, liver or pancreatic disease, colitis, constipation, 22.4%), cardiovascular diseases (high blood pressure, coronary heart disease, angina, heart attack, heart failure, 18.8%), and skin diseases (eczema, allergies, other skin rashes or varicose veins, 15.3%).

The total number of sick days recorded in the observation period was strongly correlated with age (Pearson’s r = 0.269, *p* < 0.001). Over the three-year period, no differences were observed in the number of days of absence among the different categories of workers (ANOVA F = 0.256, *p* = 0.91). Furthermore, no significant difference was found in the number of days of sickness absence in male (33.6 ± 45.6) and female employees (30.6 ± 36.5) (Student’s *t* = 0.544, *p* = 0.59).

In a simple linear regression model that included occupational category, working night shifts, gender, age, and days of absence over the three-year period as independent variables, only the last three variables were predictors of WAI score (Table 5).

### 3.9. Reliability of Reported Sickness Absences

To verify the reliability of the answers given to the question concerning the number of absences in the previous year, we correlated the value of the corresponding WAI indicator with the days of absence in the period 2021–2023. The two quantities were correlated in a very significant way (Spearman’s rho = −0.553, *p* < 0.001).

In Table 6, we have reported the mean days of sick leave taken by workers who reported a medical diagnosis for one of the envisaged classes of disease. The longest mean absence period observed between 2021 and 2023 was in workers reporting a blood disease. This was followed by mean absences for skin and other diseases.

### 3.10. Relationship between Mental Resources and Psychiatric Diseases

It is reasonable to expect that mental resources, which are the seventh dimension on which the WAI is calculated, are not equally distributed in workers with or without mental disorders.

Our analysis of variance indicated that workers who do not have mental disorders have greater mental resources. The average mental resource score in healthy workers was 3.57 ± 0.61, significantly higher (*p* < 0.001) than the score measured in workers with a personal diagnosis (2.93 ± 0.70) and with medical diagnosis of psychiatric disease (2.83 ± 0.75) No difference was observed in resources between those who declared disorders treated by a doctor and self-diagnosed disorders; in both these categories, resources were significantly lower than in healthy people.

## 4. Discussion

Applying the WAI in a healthcare company not only provides the company doctor with an initial picture of the occupational health conditions of the employees, but it also offers useful indications concerning the use of the questionnaire in healthcare activities.

In our experience, the WAI decreases significantly as the age of the workers increases. This association was reported in most previous studies [19,20,21,22,23,24,25,26,27,28], although there is no shortage of studies that failed to observe this association [29,30]. Previous authors reported that the decrease in the WAI with ageing is strictly dependent on the type of task assigned [31]. In our sample, however, the role of age was more prevalent than that of occupational category, which nevertheless had a certain relevance. Nurses had significantly lower WAI levels than other occupational categories; this result was observed in previous studies [32] and is certainly significant due to the possible consequences of healthcare activities [33]. The worsening of the WAI with age is accompanied by an increase in absences due to illness. Other studies that observed the same phenomenon demonstrated that the worsening of the WAI is followed by an increase in absenteeism [34,35]. The WAI is a recognized predictor of absenteeism [36,37,38], especially if it is accompanied by stress [39], burnout [40], or depression [41].

The data previously reported in the literature [31,32,42] led us to expect the greater impairment of self-rated working ability in women compared to men observed in our sample. We also expected to have a greater number of absences for female employees, given that women have a lower WAI than men. Interestingly, this phenomenon did not occur, since there was no difference in the average number of days of sickness absence for males and females. This suggests that women evaluate themselves as less valid than they are in reality and men declare themselves more valid than they are in reality. Probably, socio-cultural or emotional factors are involved in the self-assessment of work ability. We have planned a further study to evaluate this possibility by analyzing, in addition to the WAI, levels of occupational stress, anxiety, and depression in male and female workers. In fact, the literature tells us that the WAI score is associated with occupational stress [43,44,45,46] and emotional factors [47,48,49]. The relationship between stress, psychiatric problems, and the WAI is complex and probably cyclical: according to the Stockholm County longitudinal study, after 4 years, compared to their colleagues, workers with a low WAI had an almost twice as high a rate of distress [50]. Accurate longitudinal studies will be needed to understand whether prolonged stress worsens WAI levels and, reciprocally, whether the reduced ability of workers exposes them to greater occupational stress, with the associated consequences of physical and mental illnesses and absenteeism.

In our sample, night workers were younger and reported greater work ability than those who did not work at night. The finding we obtained was probably the result of selective factors that determined the removal of workers with health problems from night work, while younger and more capable employees carried out this type of service willingly. Previous studies on the relationship between shift work and work ability do not contradict this hypothesis. When compared to day work, no undue adverse health effects of shift schedules were observed in German workers [51] and in the Nurses’ Early Exit Study conducted in seven European countries [52]. In a Brazilian study, impairment of the WAI was observed only in workers who performed a disproportionate number of night shifts or had a temporary employment status [53]. A more recent study has shown that employees who work a greater number of night shifts are healthier than their colleagues since they have fewer chronic musculoskeletal and cardiorespiratory pathologies, a lower level of oxidation indices (DNA adducts and telomere length), and a higher WAI [54]. Therefore, selective factors appear to be predominant in the relationship between the WAI and night work among HCWs, although other factors, such as the organization of night shifts, their quantity, the type of commitment required, the rotation method, and the duration of the working hours as well as the frequency of overload are important to evaluate the harmfulness of shift work and its nterference with biorhythms.

An issue frequently addressed in the literature is the monofactorial or multifactorial nature of the questionnaire. The authors of the WAI conceived this instrument as unitary. In fact, the calculation of a total score derived from the seven components of the WAI assumes the unidimensionality of the questionnaire. The calculation of Cronbach’s alpha is also based on this assumption. In the various studies conducted with this tool, it ranged from 0.54 to 0.83 [11,55,56,57,58,59], and was 0.73 in our own study. However, it is easy to claim that work ability is a multidimensional concept and that dividing the seven dimensions into different factors could theoretically provide better results. In previous research, the underlying structure of the WAI was studied with a variety of techniques, including sampling strategies, statistical analyses, and the administration of the questionnaire in various languages and cultural contexts. These analyses produced inconsistent findings, making it difficult to interpret the WAI. Some studies confirmed the unifactorial structure of the questionnaire [56], while others recognized the existence of two [58,60,61] or three factors [55,57,59,62]. Country-specific or language-specific variations in the factor structure of the WAI were demonstrated in a study that involved about 40,000 nurses from several European nations [11]. Furthermore, different occupational categories can cause the heterogeneity observed in the WAI factor structure results. Two psychometric studies [4,61] were concordant in recognizing two factors, one composed of indicators 1, 2, 6, and 7, the other of indicators 3, 4, and 5, referring respectively to “subjective work ability and resources”, and “ill health”. In our sample, we recognized the unifactorial nature of the questionnaire and did not obtain significant advantages by subdividing it into two factors. This result can probably be attributed to the highly sectorial nature of our investigation, which was carried out in a single healthcare facility. For many years, the company in which we conducted our study had implemented an active disability management policy according to which health problems were solved by changes in the work plan to adapt the work to the person rather than by terminating employment. This policy is not uncommon in healthcare companies. Under these conditions, the two possible factors underlying the WAI merged into a single concept: difficulty in performing as effectively as in the past. Summing up, we concluded that researchers who apply the WAI in healthcare companies can consider this tool to be unifactorial.

A phenomenon observed in our sample, and expected in other healthcare companies, was the failure to find any difference between illnesses diagnosed by a doctor and self-diagnosed diseases, both of which have a significant impact on working ability. Healthcare professionals usually have sufficient competence to diagnose their own pathologies without the need to turn to a physician for confirmation; however, since the instructions of the authors of the questionnaire indicate that only diagnoses made by a doctor contribute to the formation of the final score, this introduces an inherent weakness in the assessment of the work ability of healthcare workers. Based on our experience, the shortened form of the questionnaire without the list of diseases (WAInodis) could be more effective than the full questionnaire because it contains fewer errors. However, a further study to compare the results of the WAI and the WAInodis with an external construct is needed to verify this hypothesis.

The shortened forms of the questionnaire (WAInodis and WAS) were shown to have the same distribution as the total score and to be strongly correlated with it. This suggests that they are acceptable substitutes for the full version and can be used when epidemiological needs require workers to be given a more limited number of questions. Numerous studies confirm the feasibility of using shortened versions to replace the WAI, at least in occupational health research and employee surveys [12,63,64,65]. We applied the one-item questionnaire in previous investigations on post-COVID-19 settings [66] and on the return to work of women with breast cancer [67], as well as in workplace health promotion activities [68].

Without carefully monitoring the responses, the online administration of the questionnaire probably favors the over-reporting of pathologies. We therefore paid particular attention to verifying the reliability of the workers’ answers by comparing responses obtained from the online questionnaire with data recorded in individual health records. Another critical aspect emerged when some workers declared themselves completely unable to work, but this condition was not confirmed in their health surveillance records. An in-depth investigation revealed that these workers suffered from transitory morbid conditions. A certain ambiguity therefore emerges in the first indicator of the WAI (also known as the WAS), which is interpreted by some as “at this precise moment” and by others as “in your current working condition”. These problems may occur mainly in the online administration of the questionnaire, while misunderstandings can be avoided if the questionnaire is administered immediately before the health check and the doctor can analyze it during the worker’s medical examination.

A further critical aspect concerned the definition of the work as being predominantly physical, psychological, or both, since this often led to disparities in the answers provided by workers who had the same job tasks. In our opinion, the mechanism devised by the authors to determine the second dimension of the WAI, i.e., attributing a different psychological and physical load according to whether the work is classified as psychologically or physically demanding, is certainly useful for comparing different work sectors, but may introduce a bias if applied to workers from the same healthcare company.

A careful analysis of the responses of the employees in this healthcare company revealed that more than half of them (53%) complained of musculoskeletal disorders that impaired their work ability. Mental disorders were reported by over 20% of workers and were associated with the greatest reduction in work ability. It is well known that both disorders are very frequent in healthcare activities, and may be correlated [69,70]. Studies show that the association of musculoskeletal problems and stress is responsible for a considerable reduction in WAI scores in healthcare workers [71]. The high prevalence of both diseases demonstrates the need to address this problem with environmental and organizational solutions, rather than with restrictions on individual fitness for work which can be decided by the occupational doctor. The association between mental resources and psychiatric disease leads us to believe that individual intervention aimed at improving resilience and coping strategies must be combined with structural and administrative ergonomic interventions to improve the quality of work. A systematic review with a meta-analysis of multilevel interventions in the workplace has shown that this type of combined action can bring about a significant improvement in work ability [72].

The present study has some limitations, mainly because it was conducted in a single healthcare company, so the results cannot be applied to other occupational sectors. To the best of our knowledge, there are no studies that have simultaneously considered the WAI, absences, and workers’ health records. Nevertheless, the method used in this company can be applied elsewhere and lead to the confirmation or improvement of our results. The online administration of the questionnaire allows for immediate assessment of the cohort’s conditions but may lead to over-reporting or inaccurate reporting by workers in the absence of the interaction with the occupational doctor that normally occurs when the questionnaire is completed just prior to a medical examination. Our findings have led to new research initiatives to clarify the impact of socio-cultural and personal factors on the WAI and its relationship with job satisfaction and work engagement. In addition, they have encouraged health interventions aimed at improving work from an ergonomic and interpersonal point of view.

## 5. Conclusions

This study confirms the efficacy of using the WAI for health surveillance since it enables the occupational health practitioner to quickly find out how workers assess their working ability and consequently to draw up a health plan that considers the workers’ handicaps and the best ways to remedy them. For HCWs, this instrument functions very well as a unitary indicator and there is no need to divide it into sub-scales. Shortened versions (without the list of diseases or limited to a single question) can be useful in epidemiological investigations, although the reporting of diseases can undoubtedly be advantageous for the physician in charge of workers’ health. When the WAI is used to study HCWs, it should be borne in mind that uncertainty may arise over doctor-diagnosed and self-diagnosed diseases, and it is therefore advisable to consider both conditions. The lower work ability reported by women is not associated with a greater number of absences due to illness. This should lead us to examine the socio-cultural and work-related factors that influence reporting in both genders. We hope that this article can contribute to the evaluation of the questionnaire and its application in healthcare activities.

## Figures and Tables

**Figure 1 ijerph-21-00349-f001:**
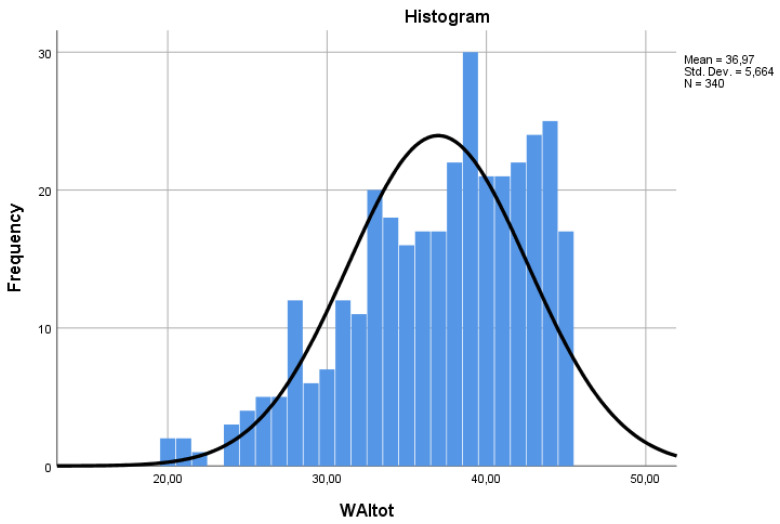
Distribution of total WAI score (WAItot).

**Figure 2 ijerph-21-00349-f002:**
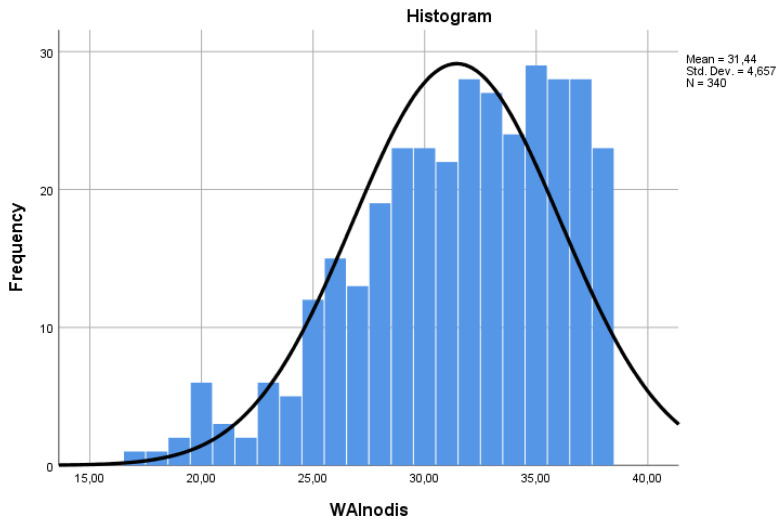
Distribution of the shortened WAI without the list of diseases (WAInodis).

**Figure 3 ijerph-21-00349-f003:**
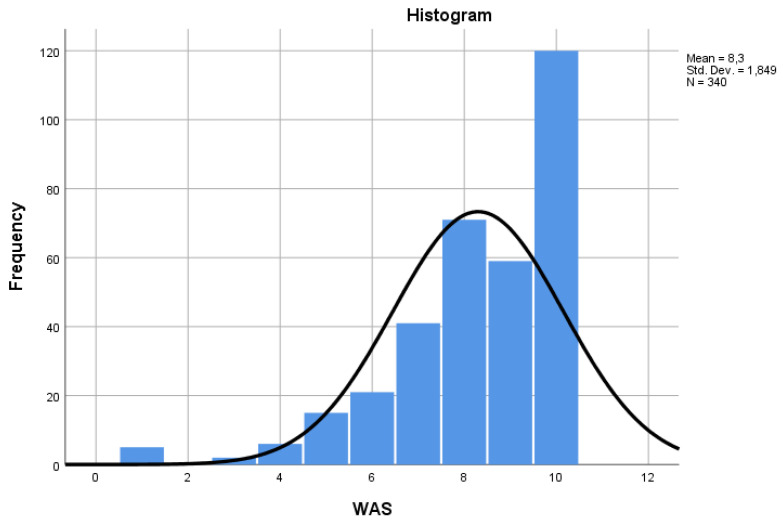
Distribution of the one-item version of the WAI (WAS).

**Table 1 ijerph-21-00349-t001:** Work ability in different categories of workers.

Category	WAI (Mean ± s.d.)	Significant Comparisons ^1^
1. Physician (n = 38)	37.1 ± 4.7	none
2. Nurse (n = 97)	35.3 ± 6.2	2 vs. 3 *p* < 0.001; 2 vs. 4 *p* < 0.01
3. Assistant nurse (n = 75)	38.8 ± 5.6	3 vs. 2 *p* < 0.001; 3 vs. 5 *p* < 0.05
4. Clerk (n = 50)	38.6 ± 4.8	4 vs.2 *p* < 0.01
5. Technician (n = 80)	36.2 ± 5.4	5 vs. 4 *p* < 0.05

^1^ Bonferroni test.

**Table 2 ijerph-21-00349-t002:** Exploratory factor analysis of the Work Ability Index administered to healthcare workers.

Items	Principal Component Analysis	Principal Component Analysis ^1^	
	Factor 1	Factor 1	Factor 2
1. Current work ability compared with lifetime best	0.71	0.73	
2. Current work ability in relation to its demands	0.74	0.75	
3. Number of current diseases diagnosed by a physician	0.62	0.59	
4. Estimated work impairment due to disease	0.70	0.68	
5. Sick leave during the past year (12 months)	0.49		0.83
6. Own prognosis of work ability two years from now	0.55	0.71	
7. Mental resources (feelings of joy, alertness, or optimism)	0.68	0.67	
Variance of the component (%)	42.09	42.09	13.57

^1^ Rotation method: oblimin with Kaiser normalization. Rotation converged in 13 iterations.

**Table 3 ijerph-21-00349-t003:** Spearman’s bivariate correlations between the shortened WAI without the list of diseases (WAInodis), the one-item form (WAS), and the total score of the complete version (WAItot).

	WAInodis	WAS	WAI
WAInodis	1.000	0.832 **	0.958 **
WAS		1.000	0.789 **
WAI			1.000

**. Correlation is significant at the 0.01 level (2-tailed).

**Table 4 ijerph-21-00349-t004:** Distribution of work ability (shortened WAInodis version without the disease-related score) in relation to the different disease classes. Analysis of variance and Bonferroni post hoc comparisons.

Class of Disease	0 No Disease n (mean ± s.d.)	1 Personal Diagnosis n (mean ± s.d.)	2 Medical Diagnosis n (mean ± s.d.)	Anova *p*	Bonferroni *p*		
					0 vs. 1	0 vs. 2	1 vs. 2
Injury	256 (32.5 ± 4.0)	49 (27.9 ± 5.0)	35 (28.7 ± 4.8)	<0.001	<0.001	<0.001	n.s.
Musculoskeletal	160 (33.8 ± 3.4)	82 (28.4 ± 4.7)	98 (30.1 ± 4.5)	<0.001	<0.001	<0.001	<0.05
Cardiovascular	276 (32.0 ± 4.3)	51 (29.1 ± 5.2)	13 (28.9 ± 5.5)	<0.001	<0.001	<0.05	n.s.
Respiratory	312 (31.7 ± 4.5)	17 (28.2 ± 5.1)	11 (30.1 ± 6.1)	<0.01	<0.01	n.s.	n.s.
Psychiatric	267 (32.7 ± 3.7)	15 (27.9 ± 5.1)	58 (26.5 ± 4.7)	<0.001	<0.001	<0.001	n.s.
Neurological	271 (32.4 ± 4.2)	32 (28.1 ± 4.5)	37 (27.3 ± 4.4)	<0.001	<0.001	<0.001	n.s.
Gastroenteric	264 (32.1 ± 4.5)	38 (29.5 ± 4.3)	38 (28.9 ± 4.7)	<0.001	<0.01	<0.001	n.s.
Genitourinary	302 (31.7 ± 4.7)	24 (29.8 ± 4.0)	14 (29.3 ± 4.6)	<0.05	n.s.	n.s.	n.s.
Skin	288 (31.9 ± 4.4)	28 (28.5 ± 4.7)	24 (28.6 ± 5.3)	<0.001	<0.001	<0.01	n.s.
Cancer	324 (31.5 ± 4.7)	12 (30.3 ± 2.5)	4 (32.5 ± 3.4)	n.s.	n.s.	n.s.	n.s.
Endocrine	277 (31.7 ± 4.6)	48 (30.0 ± 5.1)	15 (31.4 ± 4.3)	n.s.	n.s.	n.s.	n.s.
Blood	324 (31.5 ± 4.6)	11 (30.1 ± 4.2)	5 (28.2 ± 7.8)	n.s.	n.s.	n.s.	n.s.
Abortion	320 (31.5 ± 4.7)	16 (29.5 ± 4.4)	4 (33.5 ± 2.5)	n.s.	n.s.	n.s.	n.s.
Other	316 (31.7 ± 4.5)	16 (27.2 ± 5.4)	8 (30.9 ± 5.9)	<0.01	<0.001	n.s.	n.s.

Note: n.s.= not significant

**Table 5 ijerph-21-00349-t005:** Effect of gender, age, and days of absence on work ability. Simple linear regression.

	Beta	t	*p*
(Constant)		16.187	0.000
Age	−0.142	−2.304	0.022
Gender	−0.123	−2.125	0.034
Night shift	0.086	1.312	0.191
Category	0.071	1.124	0.262
Absence	−0.259	−4.317	0.000

**Table 6 ijerph-21-00349-t006:** Days of absence of workers who indicated they had a medical diagnosis for one of the 14 classes of pathologies.

Medical Diagnosis Class of Disease	Days of Absence 2021 Range (mean ± s.d.)	Days of Absence 2022 Range (mean ± s.d.)	Days of Absence 2023 Range (mean ± s.d.)	Days of Absence 2021–2023 Range (mean ± s.d.)
Injury	0–21 (5.0 ± 5.2)	0–36 (11.0 ± 10.1)	0–31 (7.9 ± 8.3)	1–83 (23.9 ± 18.5)
Musculoskeletal	0–54 (7.5 ± 10.4)	0–120 (11.8 ± 16.8)	0–83 (9.0 ± 13.4)	1–172 (28.3 ± 31.5)
Cardiovascular	0–14 (7.0 ± 4.9)	0–33 (12.6 ± 10.5)	0–134 (2.7 ± 39.7)	1–179 (40.4 ± 49.1)
Respiratory	0–172 (21.7 ± 53.1)	0–24 (9.7 ± 8.2)	0–21 (7.4 ± 7.7)	2–196 (39.8 ± 52.3)
Psychiatric	0–172 (14.8 ± 30.6)	0–64 (11.3 ± 12.6)	0–134 (15.2 ± 27.0)	1–179 (37.0 ± 47.0)
Neurological	0–36 (7.3 ± 9.2)	0–45 (13.0 ± 12.7)	0–134 (15.2 ± 27.0)	1–179 (35.5 ± 39.7)
Gastroenteric	0–172 (12.1 ± 29.7)	0–45 (11.4 ± 12.3)	0–83 (10.3 ± 17.3)	2–196 (33.8 ± 42.6)
Genitourinary	0–172 (22.4 ± 47.4)	0–40 (10.9 ± 12.7)	0–83 (10.2 ± 21.7)	1–196 (34.5 ± 60.6)
Skin	0–172 (23.2 ± 45.5)	0–45 (11.3 ± 11.0)	0–41 (8.1 ± 9.2)	2–196 (42.6 ± 48.9)
Cancer	0–10 (4.7 ± 4.2)	0–16 (7.3 ± 8.5)	2–19 (7.5 ± 7.8)	2–35 (19.5 ± 15.6)
Endocrine	0–10 (4.9 ± 4.5)	0–24 (11.0 ± 8.1)	0–66 (11.7 ± 20.7)	2–80 (27.6 ± 23.2)
Blood	0–172 (37.2 ± 75.4)	0–24 (9.2 ± 10.0)	0–19 (7.2 ± 7.6)	2–196 (53.6 ± 80.6)
Abortion	0–6 (2.2 ± 2.8)	0–13 (3.2 ± 6.5)	2–19 (7.2 ± 7.9)	2–35 (12.7 ± 15.4)
Other	0–172 (29.6 ± 63.4)	0–24 (7.0 ± 8.8)	0–19 (6.0 ± 6.6)	2–196 (42.6 ± 69.2)

## Data Availability

Anonymized data can be obtained upon request.

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
