# Peer review of "The Work Ability Index (WAI) in the Healthcare Sector: A Cross-Sectional/Retrospective Assessment of the Questionnaire"

_ijerph, 2024, doi:10.3390/ijerph21030349_

Round 1
Reviewer 1 Report
Comments and Suggestions for Authors
Dear author,
Thank you for the efforts of the respected author in writing the manuscript. A few points are raised, please note and correct:
1. The problem statement is well written.
2. According to the questions and hypotheses, the authors should explain why they did not design and develop a new tool with a more favorable score?.
3. It is also the main questionnaire for the industry. Is this questionnaire suitable for health care centers as well?
4. Why confirmatory factor analysis has not been done in addition to exploratory.
5. How to determine the sample size and sample selection should be completed with more explanations.
6. What is meant by Instrument? Don't you mean the design and development of a new tool (questionnaire)?
7. Inclusion and Exclusion criteria are not specified. These criteria should be stated more clearly according to the purpose of the study.
8. The abstract needs to be organized and completed.
9. It is better to state the exact definitions of the main variables of the study (by mentioning valid references).
Author Response
Reviewer #1
Thank you for the efforts of the respected author in writing the manuscript. A few points are raised, please note and correct:
- The problem statement is well written.
Response: We sincerely thank the reviewer for her/his appreciation of the work we have done.
- According to the questions and hypotheses, the authors should explain why they did not design and develop a new tool with a more favorable score?
R.: The reviewer kindly appreciated the aim of our work, which was to improve knowledge of the most used questionnaire in occupational medicine to measure work ability. Building a new tool to replace what has been the standard in the assessment of work capacity for over 25 years was not among the objectives of this study.
- It is also the main questionnaire for the industry. Is this questionnaire suitable for health care centers as well?
R.: The reviewer correctly observed that this tool was designed for all types of work. As we said in line 44, health care centers are one of the workplaces most frequently investigated with WAI.
- Why confirmatory factor analysis has not been done in addition to exploratory.
R.: The validity of the questionnaire has been confirmed by several psychometric investigations, conducted by the inventors of the questionnaire and by research groups in other parts of the world. This served to validate the tool and its translations into different languages. Our aim, as mentioned above, was not to develop a new questionnaire, but only to verify whether, in a situation like that of the hospital studied, the division of the score into several components which has been suggested by some authors in other contexts, proved to be effective. For this purpose an exploratory analysis is sufficient.
- How to determine the sample size and sample selection should be completed with more
R.: We gladly accepted the reviewer's suggestion, better explaining the evaluation of the sample size. Being a census, determination of minimum sample size was not necessary. The only criterion for admission was to be an active worker registered in the health surveillance registers, while the exclusion criterion was refusal to respond or incomplete response.
- What is meant by Instrument? Don't you mean the design and development of a new tool(questionnaire)?
R.: We used the term "instrument" as a synonym for "tool". The objective of developing a new questionnaire (tool or instrument) was not among ours.
- Inclusion and Exclusion criteria are not specified. These criteria should be stated more clearly according to the purpose of the study.
R.: We thank the reviewer for highlighting this point. In the "Population" subsection of the materials and methods (lines 130-134) we pointed out that in Italy workers exposed to professional risks are compulsorily subjected to health surveillance. On line 157 we wrote that "All workers exposed to occupational risks and included in the health surveillance register (575) were confidentially contacted by email and invited to participate in the survey." Therefore, our investigation was a census of all workers exposed to risk. Exclusion was motivated only by refusal to participate (line 140: Participation was completely voluntary, no economic incentive was provided for response, and no sanctions were adopted for non-responders), or incomplete responses (line 203: "62 of they were excluded because their responses were incomplete").
- The abstract needs to be organized and completed.
R.: According to editorial rules, the abstract must be a single paragraph of about 200 words maximum. The abstract must follow a structured style, but without headings: (1) Background; (2) Methods; (3) Results:; (4) Conclusions. We followed these guidelines. However, to complete the essential concepts, we had to exceed the dimensions indicated by the journal; our abstract is 248 words.
- It is better to state the exact definitions of the main variables of the study (by mentioning valid references).
R.: In this methodological study we examined the Work Ability Index and its reduced versions, sick leave and health records. We have checked the presence of definitions and references relating to the WAI.
Reviewer 2 Report
Comments and Suggestions for Authors
Dear Editor,
I would like to thank the authors of the manuscript ID: ijerph-2896582 entitled “The Work Ability Index (WAI) in the healthcare sector. A cross- sectional/retrospective assessment of the questionnaire.” for presenting the results of their study on the characteristics of different forms of WAI in a population of workers in a healthcare company and the relationship between WAI features and objective workers’ health data (age, gender, occupational and non-occupational diseases, sickness absence)
The manuscript presents the results of a cross section study that included 340 workers from a healthcare company who provided data on their name, age, gender, occupational category, night shifts, and completed the WAI questionnaire using the SurveyMonkey© online platform. The authors had access to personal health records of the participants, as well as to company sickness absence registries, which were used for obtaining objective health data for comparison. The authors used basic and advanced statistical analysis in order to check the WAI the psychometric properties (internal consistency, convergent and concurrent validity, construct validity) and to evaluate the relationship between the independent variables (gender, age, job category, night shifts, days of absence) and the WAI score.
After reading this article in details, my main impression is that the article is well written, in adherence to Journal’s standards, and addresses a very important issue of adequacy of work ability assessment in the healthcare sector using different versions of WAI. Also, the impression is that the manuscript is too long and could be possibly shortened significantly.
There are several issues for consideration:
1. Title: Adequate. No Remarks
2. Abstract: Adequate. No remarks
3. Introduction: Adequate. Remarks:
a. Despite the amount of research conducted with this tool, few critical evaluations have associated the administration of the questionnaire with objective data (e.g., the registration of absences due to illness or clinical assessments of workers) to verify the correspondence between subjective answers and objective information – References missing
b. Moreover, although several authors have used shorter versions 64 of the questionnaire (e.g., without a list of diseases or with just one component), there are 65 very few studies that have verified the usefulness of such tools – References missing
c. The WAI includes a list of diseases and seven occupational health characteristics. The final WAI score is calculated as the total unweighted score covering the WAI’s seven dimensions, or indicators. These are: • I. Current work ability compared with lifetime best; • II. Work ability in relation to the demands of the job; III. Number of current diseases diagnosed by a physician; IV. Estimated work impairment due to illness – This belongs to Methodology not to Introduction
a. To obtain a more thorough knowledge of the questionnaire, the questions we wanted to…- There are 10 separate questions presented in a numerical order taking too many lines. Is there an option to rewrite these questions in a narrative way in couple of sentences?
4. Materials and Methods: Adequate. Remarks
a. Why are the authors using a vague expression such as workers of a healthcare company? What kind of healthcare facility was analyzed?
b. …and to state whether they worked night shifts – How did the authors define night shifts, 8h vs 12h, 12-24-12-48 shift rotation? – Important for WAI analysis
c. For the period 2021 to October 2023, the healthcare company administration provided us with the absence registers for all their employees, including those who freely chose not to participate in the survey. It was the I.M.'s task to reconcile the absences with the employees who responded to the questionnaire. – Why and how did the authors access the data of patients that chose not to participate in the study? Did those patients agree to that? Who is IM and why is this a part of the Methodology: if IM is one of the Authors, this goes to Author contribution section
d. However, for non-normally distributed variables, we preferred to use non-parametric tests since, as reported by Lumley et al. [12], the assumption of normality is only required for small samples, due to the central limit theorem. With sample sizes exceeding 30 (as occurred in our case), violations of the normality assumptions are not a problem, and they become less so as the sample size increases, even when they are extreme. – Unnecessary
5. Results: Adequate. Remarks
a. What were the identified occupational risks of workers in a healthcare company?
b. Figure 2 duplicates the info given in the passage above
c. What was the gender distribution according to occupational categories? Were the gender differences in WAI persistent across all occupational categories?
d. 3.6. Reliability of the reported diseases – Too long, unnecessary details
e. Table 7. Redundant, the authors could include the findings in the text 3.10. Relationship between mental resources and psychiatric diseases
6. Discussion: remarks:
a. Line 410. We also expected to have a greater number of absences for females, given that women have a higher WAI than men. Women had lower WAI then men
b. Line 413. This finding suggests that socio-cultural or emotional factors are involved in the self-assessment of work ability. How did the authors reach this conclusion? Explain better.
c. We have planned a further study to evaluate this possibility by analyzing, in addition to the WAI, levels of occupational stress, anxiety and depression in male and female workers. In fact, the literature tells us that the WAI score is associated with occupational stress [39-42] and emotional factors [43-45]. The relationship between stress, psychiatric problems and the WAI is complex and probably cyclical: according to the Stockholm County longitudinal study, after 4 years, compared to their colleagues, workers with a low WAI had an almost two-fold rate of distress [46]. Significance for this paper?
d. The possible explanation for higher WAI in night shift workers, except the healthy worker effect, could be the type of shift (8 vs 12 hours, rotating shifts, etc), level of night strain, type of patients…The authors did not provide data on type of work, patients, shift rotation.
e. Limitations adequate
7. References: Adequate, relevant
8. Figures and tables: Adequate. Remarks mentioned
Author Response
Reviewer #2
Dear Editor,
I would like to thank the authors of the manuscript ID: ijerph-2896582 entitled “The Work Ability Index (WAI) in the healthcare sector. A cross- sectional/retrospective assessment of the questionnaire.” for presenting the results of their study on the characteristics of different forms of WAI in a population of workers in a healthcare company and the relationship between WAI features and objective workers’ health data (age, gender, occupational and non-occupational diseases, sickness absence)
The manuscript presents the results of a cross section study that included 340 workers from a healthcare company who provided data on their name, age, gender, occupational category, night shifts, and completed the WAI questionnaire using the SurveyMonkey© online platform. The authors had access to personal health records of the participants, as well as to company sickness absence registries, which were used for obtaining objective health data for comparison. The authors used basic and advanced statistical analysis in order to check the WAI the psychometric properties (internal consistency, convergent and concurrent validity, construct validity) and to evaluate the relationship between the independent variables (gender, age, job category, night shifts, days of absence) and the WAI score.
After reading this article in details, my main impression is that the article is well written, in adherence to Journal’s standards, and addresses a very important issue of adequacy of work ability assessment in the healthcare sector using different versions of WAI. Also, the impression is that the manuscript is too long and could be possibly shortened significantly.
Response: We thank the reviewer for the attention with which he examined our work and for the appreciation expressed. The suggestions were very helpful in improving the manuscript. Taking the reviewer's advice, we have reduced the scope of the article where indicated.
There are several issues for consideration:
- Title: Adequate. No Remarks
- Abstract: Adequate. No remarks
- Introduction: Adequate. Remarks:
- Despite the amount of research conducted with this tool, few critical evaluations have associated the administration of the questionnaire with objective data (e.g., the registration of absences due to illness or clinical assessments of workers) to verify the correspondence between subjective answers and objective information – References missing
Response: Following the reviewer's indications, we indicated some studies in which the WAI was correlated to the absences, or to the clinical status of the patients. However, none of these studies were intended to test the efficiency of the WAI, which is what we were discussing. In the previous version of the manuscript we had already cited in the Discussion many studies correlating WAI with absences or with disease.
- Moreover, although several authors have used shorter versions 64 of the questionnaire (e.g., without a list of diseases or with just one component), there are 65 very few studies that have verified the usefulness of such tools – References missing
R.: Several studies conducted with reduced versions of the WAI are reported in the Discussion; some of these studies evaluated the correspondence of these versions with the score obtained from the entire questionnaire. It was not adequate for us to discuss these studies in the Introduction. To accommodate the reviewer's indication we have indicated in the Introduction one of the first studies that dealt with the correspondence between the reduced forms of the WAI and the total score. We have left references to other studies dealing with reduced forms of the WAI in the Discussion.
- The WAI includes a list of diseases and seven occupational health characteristics. The final WAI score is calculated as the total unweighted score covering the WAI’s seven dimensions, or indicators. These are: • I. Current work ability compared with lifetime best; • II. Work ability in relation to the demands of the job; III. Number of current diseases diagnosed by a physician; IV. Estimated work impairment due to illness – This belongs to Methodology not to Introduction
Response: The auditor correctly observes that in surveys the description of the tools used must be made in the "Material and methods" section. This study, however, is a methodological analysis in which the WAI is the subject, not the instrument of the research. For this reason, we described the characteristics of the WAI in the Introduction, before developing research points based on these characteristics in the Introduction itself.
- To obtain a more thorough knowledge of the questionnaire, the questions we wanted to…- There are 10 separate questions presented in a numerical order taking too many lines. Is there an option to rewrite these questions in a narrative way in couple of sentences?
R.: We understand the need for synthesis expressed by the reviewer, and we know the need to contain the introduction in a reduced number of sentences. However, we know that many readers do not have in-depth knowledge of the WAI and we fear that too much summary will make it difficult to understand. The sections listed in numerical order were followed in the results paragraphs and their discussion. This certainly makes the manuscript long, but easy. However, we have tried to cut out all repetitions and unnecessary phrases throughout the manuscript.
- Materials and Methods: Adequate. Remarks
- Why are the authors using a vague expression such as workers of a healthcare company? What kind of healthcare facility was analyzed?
R.: Accepting the reviewer's indication to be more concise, we replaced the phrase with “HCWs from a public healthcare service”. The healthcare company (which is identified by the affiliation of one of the authors) is a non-state body that carries out a public hospital service.
- …and to state whether they worked night shifts – How did the authors define night shifts, 8h vs 12h, 12-24-12-48 shift rotation? – Important for WAI analysis
R.: According to Italian law, a night worker is anyone who carries out part of his working hours in the period of at least seven consecutive hours including the interval between midnight and five in the morning. As the reviewer correctly indicates, the frequency, duration and rotation of night shifts are very important for the health and well-being of workers; this type of investigation, however, goes beyond the scope of this article, which is limited to classifying the presence or absence of night shifts.
- For the period 2021 to October 2023, the healthcare company administration provided us with the absence registers for all their employees, including those who freely chose not to participate in the survey. It was the I.M.'s task to reconcile the absences with the employees who responded to the questionnaire. – Why and how did the authors access the data of patients that chose not to participate in the study? Did those patients agree to that? Who is IM and why is this a part of the Methodology: if IM is one of the Authors, this goes to Author contribution section
- The reviewer explained to us that the text was not clear. As regards this article, the company provided data on the absences of those who participated in the survey. Data on absences regularly flow to the health surveillance service, because according to Italian law, workers who are absent for more than 60 days must undergo a new medical examination before returning to work. The contribution of Dr. IM is reported in the appropriate section and consequently we have modified the text by removing this part.
- However, for non-normally distributed variables, we preferred to use non-parametric tests since, as reported by Lumley et al. [12], the assumption of normality is only required for small samples, due to the central limit theorem. With sample sizes exceeding 30 (as occurred in our case), violations of the normality assumptions are not a problem, and they become less so as the sample size increases, even when they are extreme. – Unnecessary
R.: We thank the reviewer because this comment allowed us to reduce the length of the text.
- Results: Adequate. Remarks
- What were the identified occupational risks of workers in a healthcare company?
R.: According to European directives and national law, all employers are obliged to assess the risks, with the advice of a technician (responsible for prevention and protection) and an occupational doctor (competent doctor), and to record the results of the assessment in a document. The risk assessment process can be very complex in a company. It is a continuous process, which produces individualized risk profiles for each worker. The main risks that require surveillance in a healthcare company are: biological, chemical, carcinogenic risk, radiation, load handling, night work, electromagnetic fields. Moreover, there are injury risks that do not require periodic visits, but are reported to the doctor, for example violence at work and occupational stress. Since the reviewer recognized that this article is rather long, we do not think it is appropriate to summarize the issue of occupational risks at this point. For the purposes of the article, we only wanted to point out that there is a list of workers to be subjected to health surveillance because they are exposed to one or more risks, and they have all been invited. We have changed the sentence, so that it is clearer.
- Figure 2 duplicates the info given in the passage above
R.: As the reviewer indicated, Figure 2 can be eliminated
- What was the gender distribution according to occupational categories? Were the gender differences in WAI persistent across all occupational categories?
- By dividing the sample into 5 components corresponding to the different professional categories, the difference between males and females is no longer statistically significant.
- 3.6. Reliability of the reported diseases – Too long, unnecessary details
R.: We summarized the text, sacrificing some details and eliminating nine lines.
- Table 7. Redundant, the authors could include the findings in the text 3.10. Relationship between mental resources and psychiatric diseases
R.: Following the reviewer's indication, we eliminated Table 7, inserting the results into the text.
- Discussion: remarks:
- Line 410. We also expected to have a greater number of absences for females, given that women have a higher WAI than men. Women had lower WAI then men
R.: We thank the reviewer who reported us a “lapsus calami”. We replaced higher with lower.
- Line 413. This finding suggests that socio-cultural or emotional factors are involved in the self-assessment of work ability. How did the authors reach this conclusion? Explain better.
R.: What is indicated here is not a conclusion, but a hypothesis. Considering the inverse relationship between WAI and absences, and the fact that women have a lower WAI than men, they should have more absences than men. If this does not happen, it can be thought that women evaluate themselves as less valid than reality and men declare themselves more valid than reality. We have detailed this hypothesis.
- We have planned a further study to evaluate this possibility by analyzing, in addition to the WAI, levels of occupational stress, anxiety and depression in male and female workers. In fact, the literature tells us that the WAI score is associated with occupational stress [39-42] and emotional factors [43-45]. The relationship between stress, psychiatric problems and the WAI is complex and probably cyclical: according to the Stockholm County longitudinal study, after 4 years, compared to their colleagues, workers with a low WAI had an almost two-fold rate of distress [46]. Significance for this paper?
R.: We have added as an explanation the notation that: accurate longitudinal studies will be needed to understand whether prolonged stress worsens WAI and, reciprocally, the reduced ability of workers exposes them to greater occupational stress, with the associated consequences of physical and mental illnesses and absenteeism. It seems to us that the concept is now clearer; We thank the reviewer who allowed us to explain better.
- The possible explanation for higher WAI in night shift workers, except the healthy worker effect, could be the type of shift (8 vs 12 hours, rotating shifts, etc), level of night strain, type of patients…The authors did not provide data on type of work, patients, shift rotation.
R.: The reviewer is absolutely right. The organization of night shifts, their quantity, the type of commitment required, the rotation method and the duration of the working hours as well as the frequency of overload are all very important factors in evaluating the harmfulness of night work. Our study group is currently engaged in studies on this topic in two different projects in collaboration with other groups of scholars. However, in this research the data was not collected. We have added a notation to remember what the reviewer revealed.
- Limitations adequate
- References: Adequate, relevant
- Figures and tables: Adequate. Remarks mentioned
Reviewer 3 Report
Comments and Suggestions for Authors
Work Ability is a very important research topic. However, this study only repeatedly investigated various facts already known about Work Ability, and no new findings were identified, so I am not sure what kind of scientific contribution this study can make compared to previous studies. I think it would have been better to specify the research topic and focus on it to write the paper.
Author Response
Reviewer #3
Work Ability is a very important research topic. However, this study only repeatedly investigated various facts already known about Work Ability, and no new findings were identified, so I am not sure what kind of scientific contribution this study can make compared to previous studies. I think it would have been better to specify the research topic and focus on it to write the paper.
Response: We thank the reviewer for taking the time to read our work. The comments we received, although very brief (65 words), allowed us to improve the text, underlining the progress that our study has made compared to previous literature, which was very rich in contributions but less rich in methodological analyses. To the best of our knowledge, there are no studies that have simultaneously considered WAI, absences, and workers' health records. We also revised the 10 topics we listed in the Introduction, to ensure that every reader could recognize the progression throughout the manuscript.